# Selection of Suppliers for Speech Recognition Products in IT Projects by Combining Techniques with an Integrated Fuzzy MCDM

**Atour Taghipour** [1], **Babak Daneshvar Rouyendegh** [2,*], **Aylin Ünal** [2] **and Sujan Piya** [3]

1 Faculty of International Business, Normandy University, 76600 Le Havre, France;
atour.taghipour@univ-lehavre.fr
2 Department of Industrial Engineering, Ankara Yıldırım Beyazıt University, Ankara 06010, Turkey;
aylinunal02@gmail.com
3 Department of Mechanical and Industrial Engineering, College of engineering, Sultan Qaboos University,
Muscat 123, Oman; sujan@squ.edu.om
* Correspondence: atour.babek.erdebilli2015@gmail.com

**Abstract:** In today's environment, as the complexity of actual events develops, products become increasingly complicated. As a result, companies should collaborate to integrate disparate technologies while developing a product or service. Additionally, collaborating with the right supplier helps a company increase the flexibility, competitiveness, and profitability of its goods or services. The goal of this study is to look into the factors that influence supplier selection for speech recognition. Twelve sub-criteria for quality, affordability, maintenance, and adaptability are used to evaluate prospective providers. Two separate hybrid methodologies for finding the best supplier of an information technology product are presented. intuitionistic Fuzzy Due to the uncertainty of the data, VIKOR operates as the decision-making matrix and solves the issue by determining the ideal alternative for group utility using VIKOR. The second technique, Q-ROF TOPSIS, selects suppliers by utilizing q-rung orthopair fuzzy sets, which provides decision makers with greater expression flexibility than the majority of uncertainty-related strategies. To demonstrate the effectiveness of the recommended measures, a case study is conducted. The outcomes of various strategies are compared, as well as the associated advantages.

**Keywords:** integrated intuitionistic Fuzzy MCDM; Q-ROF TOPSIS; speech recognition; supplier selection

## 1. Introduction

Today, the world is attempting to contain the COVID-19 outbreak. Individuals' day-to-day functioning has been impacted by the epidemic. Additionally, individuals, organizations, and enterprises are looking for novel ways to conduct operations remotely. This necessitates the development of new technologies that assist them in successfully sustaining their technique. The case study in this research is a contact center firm. The company's bots and Interactive Voice Responses (IVRs) are meant to help companies improve customer loyalty and satisfaction while simultaneously cutting contact center operational expenses. Customer experience has become more crucial than ever for almost all companies as a result of new technology and a changing environment. Nowadays, chatbots, interactive voice response systems, and mobile assistants are commonly used to increase the customer experience across virtually all industries, including healthcare, banking, government, and insurance. All of these bots and IVRs require a certain level of technology to operate normally. At this level, speech recognition (SR) is crucial for converting speech to text, allowing a bot, interactive voice response system, or mobile assistant to comprehend what the user says along this journey. Choosing the right supplier of SR technology is critical

since it enables the organization to enhance the customer experience by delivering an easy-to-manage and build conversational system. Another purpose for having the best SR technology is to let consumers to operate products with minimal effort. Not just for individual users, it simplifies the process of trancribing massive obligations incurred in sectors such as healthcare, insurance, and banking. Additionally, true SR technology is critical for organizations' automation, which includes transforming complex IVR menus into customer-friendly systems, enhancing security, providing an alternative to touchtone menus for customers who prefer not to use touchtone menus, and lowering operational costs. The technology's popularity has risen dramatically in recent years as a result of market requirements and opportunities. Choosing the appropriate provider of SR technology on a global scale has become a difficult process, since there are numerous vendors and numerous aims to accomplish. This study examined four suppliers on the basis of four primary criteria and twelve sub-criteria. These providers were chosen as the most sought-after suppliers of SR technology on the market. The criteria for choosing the best supplier from a group of four are based on the most critical aspects of all supplier's products, the opinions of authorized people, and customer feedback. The essential characteristics addressed are quality, price, maintenance, and adaptability. The quality of SR technology is defined by the system's accuracy in translating speech to text. A better level of precision translates into more precise automation for IVRs, medical assistants, and so on. For instance, if a speech recognition engine incorrectly recognizes a phrase, subsequent components may incorrectly recognize the user intent. It results in a negative customer experience and increased process costs due to a lack of automation. As a result, accuracy is used to quantify quality. Additionally, unit price and other costs are critical criterion for picking a supplier, as one of the constant constraints on businesses is budget. Additionally, maintenance is critical for developing long-term customer service systems. As a result, all system components must be maintained optimally and correctly. Finally, flexibility is critical, because even if a voice recognition system is the highest quality, the cheapest, or the easiest to maintain, it is not acceptable for a business if the provider does not support adequate integration or installation solutions. This investigation makes use of many criteria with finite options. Each selection includes qualitative and quantitative data, as well as details about the criterion and sub-criteria. Given these characteristics of the problem, VIKOR is chosen as the Multi Criteria Decision Making (MCDM) approach; however, to account for the effect of data uncertainty, this research incorporates intuitionistic fuzzy (IF) into VIKOR. Based on the research conducted by Kabak and Çınar [1] in their book "Yönetimde çok kriterli karar verme yöntemleri," VIKOR enables the management of numerous qualitative and quantitative factors in order to arrive at the optimal solution. VIKOR was chosen as the MCDM approach because it includes both positive and negative criteria for maximizing social benefit while minimizing individual regret. The IF approach assists in mitigating the influence of ambiguity in concepts that rely on decision makers' judgment, whereas VIKOR ranks the possible outcomes and selects the closest to optimal outcome. Additionally, because of its parametric character and susceptibility to uncertainty, we use q-ROF TOPSIS as a secondary approach. Finally, we compare the two strategies' outcomes and demonstrate their respective benefits. The contributions of this study are as follows:

➢ qualitative criteria for IT vendor selection are utilized in conjunction with quantitative criteria to evaluate speech recognition software; and
➢ the suggested technique quantifies verbal expressions and reduces ambiguity in decision making.

Although the necessity for automation is a very common topic because to COVID-19 and technology, the amount of study conducted across multiple disciplines is quite limited. Generally, IT research in the literature focuses exclusively on the solution's unique criteria. Additionally, these criterion are predominantly quantitative data in the literature. For instance, the most often used criterion in the literature for comparing voice recognition providers is the word-of-error rate. However, there are numerous criteria for selecting the most suited source as a voice recognition supplier. Not only should the best provider be

the highest quality or the cheapest, but it should also be the most adaptable and simple to use. Due to the uncertainty, qualitative data is critical as a criterion. As a result of this predicament, numerous supporter selection studies use not only quantitative but also qualitative data. On the other hand, there is a void in the literature for supplier selection problems in the IT industry using these criteria, despite the fact that research in the supplier selection field continues to grow at a rapid pace.

Numerous qualitative and quantitative criteria are evaluated in this study in order to eliminate uncertainty and provide a fresh perspective on the problem of supplier selection in the information technology industry.

The remainder of the investigation is as follows. Section 2 reviews the literature. The challenge addressed by this study is defined in Section 3 of the structural problem. Section 4 discusses the IF-VIKOR and q-ROF TOPSIS methodologies. Section 5 presents a case study of supplier selection and resolves it using two ways. Section 6 discusses the case study's findings, while Section 7 outlines many conclusions and future research directions.

## 2. Literature Review

Supplier selection has been a significant study issue for a long period of time, and several studies have presented various strategies for selecting the best supplier depending on research topics and data types in the past [2–4]. Although the majority of studies employed AHP, ANP, TOPSIS, and their combinations with the fuzzy set to choose the best supplier, VIKOR, DEMATEL, and ANOVA were also used often. VIKOR has been employed in over 750 supply chain management (SCM) studies over the last two decades, and there are over 500 researches on the intuitionistic fuzzy approach to SCM challenges [5,6]. According to research, VIKOR is typically used in conjunction with fuzzy AHP, fuzzy ANP linguistic information, fuzzy DEMATEL, ELECTRE, and BWM, and other fuzzy techniques, whereas intuitionistic fuzzy is typically used in conjunction with MULTIMOORA, AHP, PROMETHEE, UTASTAR, ELECTRE, and ANP.

Qun et al. [7] solved a green supplier selection issue by integrating VIKOR and BWM in an interval type-2 fuzzy environment. They profited from the VIKOR approach for making multi-criteria decisions as well as the best-worst method for removing subjectivity. Awashi et al. [8] solved a multi-tier supplier selection issue by combining VIKOR and fuzzy AHP techniques. The study defines the weights of criteria using fuzzy AHP and evaluates vendors using fuzzy VIKOR. Rouyendegh et al. [9] examined the performance of suppliers using a hybrid AHP-IFT model. When using AHP as an MCDM approach, they rank decision makers (DMs) using intuitionistic fuzzy numbers. Rouyendegh et al. [10] suggested an intuitionistic and fuzzy TOPSIS technique for solving the green supplier selection problem. IF is used in this study to reduce subjectivity in data collection from DMs. Pınar and Boran [11] suggested a strategy for green supplier selection based on a q-rung orthopair fuzzy TOPSIS. Chen and Wang [12] confirmed fuzzy VIKOR using a systematic and reasonable method for identifying partner problems in information systems and information technology projects. Rouyendegh [13] validated a hybrid technique for supplier selection issues involving uncertainty and subjectivity by combining ANP and IF sets. Çalı and Balaman [14] recommended integrating VIKOR and ELECTRE 1 in order to utilize supplier assessment in an unclear area. Kumar et al. [15] examine how integrated AHP, TOPSIS, and Taguchi loss functions can be used to solve the supplier selection problem for India's heavy locomotive manufacturer. The problem specifies quality, lead, and cost as requirements. Baset et al. [16] proposed using a hybrid Neutrosophic ANP and VIKOR to choose a suitable provider in an uncertain environment and with limited knowledge. ANP is used to calculate the weights of the main and sub-criteria, and then VIKOR is used to select the optimum solution. Kamalakannan et al. [17] suggested that a TOPSIS approach be used to solve the supplier selection problem, taking into consideration not only profit but also green management. Memari et al. [18] describe a sustainable supplier selection problem with multi-criteria problem is solved using intuitionistic fuzzy TOPSIS technique. There are nine primary, and thirty sub-criteria are evaluated in the problem.

The study examines supplier outranking using a Multi Criteria Group Decision Making environment. The following table summarizes the results of a literature review conducted in order to rate voice recognition products. The overview of the literature review on supplier selection is shown in Table 1, along with the MCDM techniques utilized in each research. In Table 2, WER (word-error-rate) denotes the product's accuracy level, process performance denotes the product's real-time factor (speed), file format denotes the variety of file formats supported by the product, and multi-language denotes the variety of languages supported.

**Table 1.** Summary of literature review for supplier selection problems.

| Author (Year) | Title | VIKOR | Best-Worst | IF | Fuzzy AHP | ELECTRE | Fuzzy TOPSIS | TOPSIS | Fuzzy VIKOR | ANP |
|---|---|---|---|---|---|---|---|---|---|---|
| Wu Qun et al. [7] | An integrated approach to green supplier selection based on the interval type-2 fuzzy best-worst and extended VIKOR methods | √ | √ | | | | | | | |
| Awasthi Anjali et al. [8] | Multi-tier sustainable global supplier selection using a fuzzy AHP-VIKOR based approach | √ | | | √ | | | | | |
| Kumar, et al. [15] | AHP, TOPSIS and Taguchi loss function are combined to solve supplier selection problem | √ | | | √ | | | | | |
| Baset, et al. [16] | hybrid Neutrosophic ANP and VIKOR are applied together to select appropriate supplier in uncertain environment | √ | | | | | | | | √ |
| Kamalakannan, et al. [17] | TOPSIS is selected as an approach for supplier selection problem | | | | | | | √ | | |
| Memari Ashkan et al. [18] | Sustainable supplier selection: A multi-criteria intuitionistic fuzzy TOPSIS method | | | √ | | | | √ | | |
| Rouyendegh et al. [10] | Intuitionistic Fuzzy TOPSIS method for green supplier selection problem | | | √ | | | | √ | | |
| Rouyendegh, et al. [9] | An AHP-IFT Integrated Model for Performance Evaluation of E-Commerce Web Sites | | | √ | √ | | | | | |
| Rouyendegh [13] | Developing an Integrated ANP and Intuitionistic Fuzzy TOPSIS Model for Supplier Selection | | | √ | | | √ | | | √ |
| Pınar Adem, et al. [19] | q-rung orthopair fuzzy TOPSIS method for green supplier selection problem | | | | | | √ | | | |

**Table 2.** A literature review for ranking speech recognition products.

| Author | Title | WER | Process Performance | File Format | Multi-Language Speech Recognition |
|---|---|---|---|---|---|
| **Kim et al. [20]** | A Comparison of Online Automatic Speech Recognition Systems and the Nonverbal Responses to Unintelligible Speech | √ | √ | √ | |
| **Herchonvicz et.al. [21]** | A comparison of cloud-based speech recognition engines | √ | √ | | √ |
| **Bisani et.al. [22]** | Bootstrap estimates for confidence intervals in ASR performance evaluation | √ | | | |
| **Dharmani et.al. [23]** | Performance Evaluation of ASR for Isolated Words in Sindhi Language | √ | | | √ |
| **Gonzalez et al. [24]** | An Illustrated Methodology for Evaluating ASR Systems | √ | | | |

Technical comparisons of SR technology suppliers have historically emphasized quantitative criteria in the literature. Kim et al. 2019 rank five speech-to-text engine providers according to their accuracy, word-error rates, and performance. The research uses a variety of audio kinds to conduct evaluations in a variety of audio formats. Herchonvicz et al. [21] investigated several measuring methodologies for ranking engines according to their per-

formance. Different accents within a same language are also discussed in the research. Mahmoudi et al. [25] introduced fuzzy TOPSIS and OPA in a large-scale MCDM with missing data for project selection. The literature indicates that research is mostly focused on developing new algorithms or approaches to improve accuracy or on deploying voice recognition solutions to various industries. While there is research comparing the accuracy of different speech recognition engines, there is a lack of research comparing them using different criteria. The clustering requirements are defined using Principal Component Analysis, the clustering alternatives are determined using the K-Algorithm, and the clusters are ranked using fuzzy TOPSIS and OPA.

In light of the review, VIKOR is advantageous because it enables the ranking of all alternatives closest to the ideal based on the greatest group benefit and the least individual regret using not only positive but also negative criteria that incorporate intuitionistic fuzzy logic to eliminate uncertainty. Additionally, q-ROF TOPSIS approaches enable decision-makers to express themselves more freely than most other ways for dealing with uncertainty.

### 3. Structuring the Problem

In this section, an MCDM technique is used to pick a provider for a voice recognition engine. Additionally, the IF set is utilized to generate a decision matrix in order to mitigate the influence of DMs' subjective judgment. The problem is stated in this study as selecting the best voice recognition supplier for a business's IVR and Virtual Assistant solutions. Speech recognition positions are available in this solution for converting speech to text in order to automate the procedure and improve the client experience. Thanks to speech recognition, the system can comprehend what the user has said and assist in defining the next stage in the dialog flow. For instance, when customers contact the company's customer service, the IVR prompts the user, "What transaction do you wish to initiate?" The consumer then responds, "I want to pay off my credit card debt". As a result, the customer support system may direct the user to the appropriate menu without involving any human resources. In this case, speech recognition is the engine that interprets what the user says and communicates it to the other components.

There are numerous factors for selecting the best supplier for these solutions, and numerous limits obstruct all of them. As a result, it transforms into a multi-criteria decision-making problem with restricted alternatives and several objectives.

#### 3.1. Current State

According to the analysis of the literature, as indicated in Table 1, scientific measuring techniques and algorithms are typically employed to rank speech recognition engines. The primary limitation of these techniques is that they can only take into account quantitative data. As a result, this study sought to overcome this restriction by including qualitative factors into the problem. Additionally, as indicated before in the introductory part, the product is a core technology, which means it continually improves with the advancement of machine learning technologies.

#### 3.2. Suggested Improvements

Due to the gap identified in the literature analysis, this study applies the MCDM technique to the quantifiable and unmeasurable factors of the supplier selection problem for SR goods. It is supplied to account for unquantifiable characteristics, so that purchasers may be certain they did not simply address a few. As a result, this paper proposes a hybrid technique for addressing restrictions inherent in the current state of the supplier selection problem for products in the information technology industry. The literature is combed in order to choose the best appropriate MCDM approach. The supplier selection challenge is represented in Table 2 based on a survey of the literature. The table defines the most often used criteria and approaches. The methods outlined below is advised in light of studies. After defining the problem in the preceding section, the primary criteria are created

using a combination of a literature assessment of SR performance and evaluations and general supplier selection research. The purpose of merging this research is to provide a novel way to evaluating speech recognition services that takes into account existing research limits on handling. The most often used primary criteria in the literature are quality (accuracy rate), cost, maintenance, and adaptability, which are listed in Table 2. After determining the primary criteria, as seen in Figure 1, sub-criteria for these primary criteria are developed. The accuracy sub-criteria are specified as word-error rate and technical leadership. The word-error-rate is included as a sub-criterion since it is the most widely used and quantifiable metric for determining the effectiveness of speech recognition (ref. if any). Technological leadership is included as a sub-criterion since speech recognition is a critical technology that should be constantly improved with the addition of new AI and machine learning technologies. Similarly, cost sub-criteria include recognition cost (the price of a product) and hardware expenses. Because the cost of SR technology may be classified into two categories based on unit cost and hardware cost, they are included as sub-criteria under the cost in this study. These two charges are substantial since they have a direct impact on the total cost. Following that, sub-criteria for maintenance include support activities, workflow tools, the number of supported format types, and the capacity to handle all accents. Workflow tool is picked as a sub-criterion due to its ability to facilitate product integration, development, and testing. Collaboration is facilitated by sharing workflow tools with the consumer. Support activities are critical while selecting an SR supplier due to the product's maintenance in the event of an issue. Solving problems at the appropriate time is critical for enterprises to provide positive customer experiences. The number of supported formats is one of the indicators of a product's adaptability, as various AI and conversational products may demand a different sort of output or input. The ability to handle all dialects is critical when picking a provider, as it avoids additional expenditures and a lack of quality when separate language models are necessary for different accents. When transcribing a call from a financial company's contact center, for example, your representative may talk in American English while the consumer speaks in Indian English. Finally, for the flexibility criteria, numerous languages, experienced industries, integration capabilities, and ease of deployment were chosen as sub-criteria. Multiple language support and industry experience are viewed as critical features of a product by all suppliers, and hence are included as sub-criteria. Integration capabilities enables enterprises who utilize the SR product to lower their integration costs, while quick deployment enables these organizations to reduce their maintenance costs.

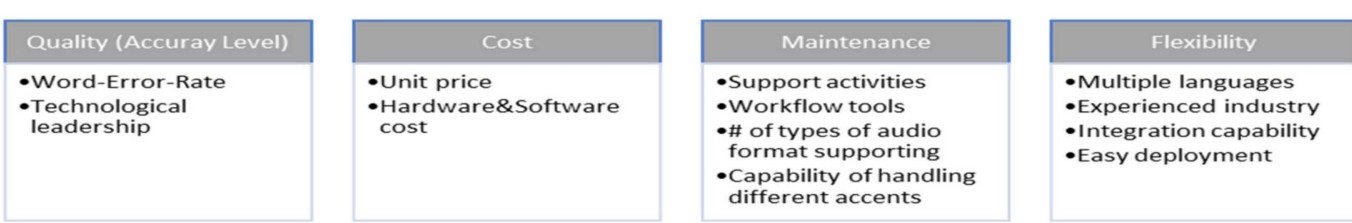

**Figure 1.** Main criteria and Sub Criteria.

Because the problem in this research contains more than one criterion with finite options, MCDM approaches can be used to solve it. VIKOR is used as the MDCM approach in our study, and owing to the presence of both quantitative and qualitative data, intuitionistic fuzzy logic is incorporated into the process to reduce ambiguity. In summary, Figure 2 depicts a systematic technique.

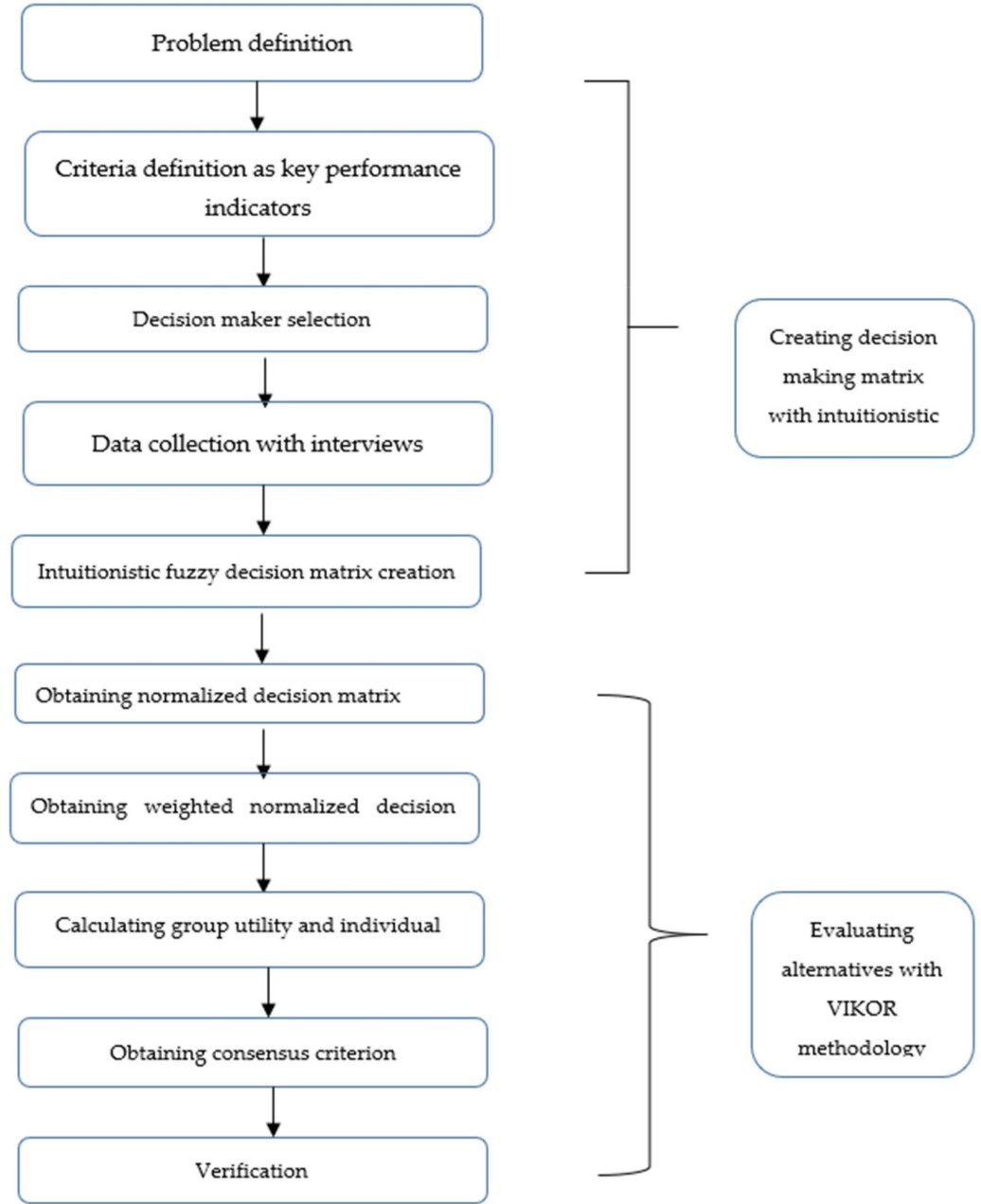

**Figure 2.** Suggested Methodology.

## 4. Constructing the Decision Model

There are quantifiable and unquantifiable factors in this study, and data is collected from DMs, therefore subjectivity should be excluded while generating the decision matrix. Comparisons with other types of MCDM approaches might be conducted to bolster the study's credibility. The study's appeal can be enhanced by combining the results of all compared techniques into a single figure. VIKOR and q-ROF are used in an uncertain environment to cover all of these data. To deal with ambiguity, an intuitive fuzzy technique is adopted. Following the creation of a decision matrix, the alternatives are rated using the VIKOR method. Additionally, a model based on the q-ROF TOPSIS is used to evaluate the performance of the approaches.

### 4.1. Intuitionistic Fuzzy Sets

Fuzzy sets provide ranking membership with grading scores from 0 to 1. Based on the study of [9], basic concepts of IF set is defined as A= {(r, $\mu_A$(r), $v_A$(r) $\vdots$ r∈R}, while $\mu_A$(r) is membership degree of function, $v_A$(r) is not membership.

In fuzzy set, summation of these functions should be greater than or equal to 0 and less than or equal to 1. Also, $\pi_A$(r) is the intuitionistic fuzzy index—belonging to *A*.

$$\pi_A(r) = 1 - [\mu_A(r) + v_A(r)] \tag{1}$$

where $\pi_A$(r) is the uncertainty of r to A for every r ∈ R

$$0 \leq \pi_A(r) \leq 1 \tag{2}$$

$\pi_A$(r) defines the degree of uncertainty. Having more knowledge of r, the number should be smaller.

$$\mu_A(r) = 1 - v_A(r) \tag{3}$$

### 4.2. q-Rung Orthopair Fuzzy Sets

After IF sets, Yager [11] introduced Pythagorean fuzzy sets, membership and non-membership degrees (*a*, *b*) such that *a*, *b* ∈ [0, 1] as follows:

$$a^2 + b^2 \leq 1 \tag{4}$$

He proposed a q-ROF subset A of X, is given as below:

$$A = \{\langle x, \mu_A(x), v_A(x)\rangle | x \in X\} \tag{5}$$

where $\mu_A : X \to [0,1]$ is membership degree and $v_A : X \to [0,1]$ is non-membership degree of $x \in X$ to the set A with the condition given below:

$$(\mu_A(x))^q + (v_A(x))^q \leq 1 \tag{6}$$

The degree of hesitancy in q-ROF sets is indicated with $\pi$ and defined as

$$\pi_A(x) = \left(1 - (\mu_A(x))^q - (v_A(x))^q\right)^{1/q} \tag{7}$$

Therefore, q-rung orthopair fuzzy set (q-ROFs) is a generalized version of IF (*q* = 1) and Pythagorean fuzzy (*q* = 2) sets.

### 4.3. IF-VIKOR

**Step 1.** To use VIKOR methodology at first it is necessary to identify goal, criteria and alternatives. As discussed before, the goal is to identify the best supplier of IT product and the criteria for the selection were elucidated in Section 3. The alternative here represents the available suppliers; Figure 3 depicts a First step of VIKOR:

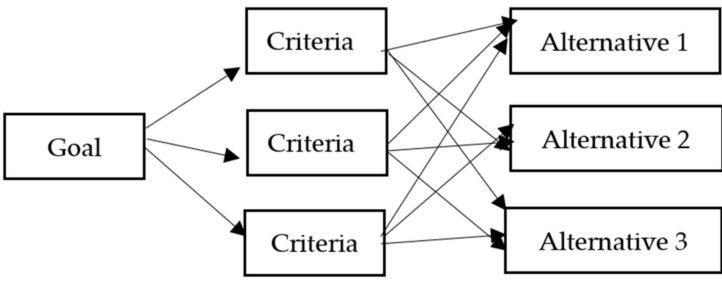

**Figure 3.** First step of VIKOR.

**Step 2.** To obtain a decision matrix using IF method for each alternative based on the defined criteria and sub-criteria. In the

**Step 3.** The decision matrix obtained in step two is normalized with the following equations of intuitionistic fuzzy methodology. Normalization is necessary to avoid the effect of different unit related to various criteria and sub-criteria.

$f_{ij}$ refers to elements of decision matrix as ith alternative and jth criteria

$r_{ij}$ refers to elements of normalized decision matrix as ith alternative and jth criteria

$$f_j^+ = max_i f_{ij} \tag{8}$$

$$f_j^- = min_i f_{ij} \tag{9}$$

$$r_{ij} = \frac{f_j^+ - f_{ij}}{f_j^+ - f_j^-} \tag{10}$$

$$R = \begin{matrix} r11 & r12 & \dots & r1n \\ r21 & \dots & & r2n \\ \vdots & \ddots & & Rmn \end{matrix}$$

**Step 4.** The decision matrix is weighted reflecting the effect levels of the criteria on the decision.

$$v_{ij} = r_{ij} w_j \tag{11}$$

$v_{ij}$ refers to elements of weigthed normalized decision matrix as ith alternative and jth criteria.

**Step 5.** Individual regret and group utilities are calculated. Group utility ($S_i$) refers to the total weighted normalized value to be obtained if the alternative is selected which means best situation, while individual regret ($R_i$) indicates the largest record that will occur based on a criterion if the alternative is not selected which means worst.

$$S_i = \sum_{j=1}^{n} W_j \frac{f_j^+ - f_{ij}}{f_j^+ - f_{ij}^-} = \sum_{j=1}^{n} V_{ij} \tag{12}$$

$$R_i = max_j (w_j \frac{f_j^+ - f_{ij}}{f_j^+ - f_{ij}^-}) = max_j V_{ij} \tag{13}$$

**Step 6.** Ranking indexes ($Q_i$) are computed. The consensus criterion is calculated to ensure that group utility and individual regret criteria are combined in order to decide between alternatives.

$$Q_i = \theta \times \frac{S_i - S^+}{S^- - S^+} + (1 - \theta) \times \frac{R_i - R^+}{R^- - R^+} \tag{14}$$

**Step 7.** the best alternatives are ranked. For verification, it is examined if the results meet the conditions below or not.

First condition: Advantage Acceptance

Suppose that the alternative with the lowest $Q_i$ value has the Q (a′) value, the second-best alternative (Q (a″)) and DQ value is defined as equal to 1/(m − 1).

It is acceptable when

$$Q (a'') - Q (a') \geq DQ \tag{15}$$

Second condition: Stability Acceptance

The choice with the best $Q_i$ value should also be the best alternative from the point of group benefit and/or individual regret criteria.

When both conditions are fulfilled, the $Q_i$ value is determined as the best alternative with compromise solution. Only if Condition 1 is satisfied, two alternatives with the

best consensus criterion value will be determined as the best solution. If Condition 1 is not verified, all alternatives up to the next alternative are determined as compromise solutions until $m^{th}$ confirms the condition based on the following expression which is met the requirement.

*4.4. q-ROF TOPSIS Method*

Let $A = \{A_1, A_2, A_3, \cdots, A_m\}$ be a set of alternatives and $X = \{X_1, X_2, X_3, \cdots X_n\}$ be a set of criteria, the modified q-ROF TOPSIS method [21] steps are given below:

**Step 1.** Aggregate the DMs ratings and obtain a decision matrix.

First, DMs evaluate the vendors with the linguistic terms. These terms are then converted to q-ROFNs. Suppose $\alpha_k = \mu_k(x), v_k(x)(k = 1, 2, 3 \cdots, l)$ is a group of q-ROF numbers which are aggregated with DM weights $(\lambda_k)$ with the help of q-ROFWA operator below [21]

$$q - ROFWA(\alpha_1, \alpha_2, \cdots, \alpha_l) = \left\langle \left( 1 - \prod_{k=1}^{l}(1 - \mu_k(x)^q)^{\lambda_k} \right), \prod_{k=1}^{l} v_k(x)^{\lambda_k} \right\rangle \quad (16)$$

q-ROF decision matrix is as follows:

$$R = \begin{bmatrix} \mu_{A_1}(x_1), v_{A_1}(x_1), \pi_{A_1}(x_1) & \mu_{A_1}(x_2), v_{A_1}(x_2), \pi_{A_1}(x_2) & \cdots & \mu_{A_1}(x_n), v_{A_1}(x_n), \pi_{A_1}(x_n) \\ \mu_{A_2}(x_1), v_{A_2}(x_1), \pi_{A_2}(x_1) & \mu_{A_2}(x_2), v_{A_2}(x_2), \pi_{A_2}(x_2) & \cdots & \mu_{A_2}(x_n), v_{A_2}(x_n), \pi_{A_2}(x_n) \\ \vdots & \vdots & \ddots & \vdots \\ \mu_{A_m}(x_1), v_{A_m}(x_1), \pi_{A_m}(x_1) & \mu_{A_m}(x_2), v_{A_m}(x_2), \pi_{A_m}(x_2) & \cdots & \mu_{A_m}(x_n), v_{A_m}(x_n), \pi_{A_m}(x_n) \end{bmatrix}$$

$R = (r_{ij})_{mxn}$ where $(\mu_{A_i}(x_j), v_{A_i}(x_j), \pi_{A_i}(x_j))$, $(i = 1, 2, \ldots, m; j = 1, 2, \ldots, n)$.

**Step 2.** Calculate the weights for the criteria.

To determine the evaluation criteria's significance degrees ($Wj$), all language words scored by DMs are translated to q-ROFNs using Equation (17):

$$W_j = \frac{\sum_{k=1}^{l} \lambda_k (1 + \mu_k^q(x_j) - v_k^q(x_j))}{\sum_{j=1}^{n} \sum_{k=1}^{l} \lambda_k (1 + \mu_k^q(x_j) - v_k^q(x_j))} \quad (17)$$

**Step 3.** Create a decision matrix that is weighted.

The aggregated weighted q-ROF decision matrix is created using the methods presented in Equation (18): [26]:

$$w_k \alpha_1 = \left\langle \left( 1 - (1 - \mu_1(x)^q)^{w_k} \right)^{1/q}, v_1(x)^{w_k} \right\rangle$$

$$\pi_{A_i}(x_j) = \left( 1 - \mu_{A_i}^q(x_j) - v_{A_i}^q(x_j) \right)^{1/q} \quad (18)$$

$r'_{ij} = (\mu'_{ij}, v'_{ij}, \pi'_{ij}) = (\mu_{A_iW}(x_j), v_{A_iW}(x_j), \pi_{A_iW}(x_j))$ is an element of the matrix where $(i = 1, 2, 3, \ldots, m; j = 1, 2, 3, \ldots, n)$.

**Step 4.** Determine the Positive and Negative Ideal Solutions:

q-ROF Positive Ideal Solution (q-ROFPIS, $A^*$) maximizes the benefit and minimizes the cost, contrarily, q-ROF Negative Ideal Solution (q-ROFNIS, $A^-$) minimizes the benefit and maximizes the cost. So, let $J_1$ and $J_2$, be benefit and cost criteria respectively. $A^*$ (q-ROFPIS) and $A^-$ (q-ROFNIS) is obtained with the below formula:

$$A* = (\mu_{A*W}(x_j), v_{A*W}(x_j)) \text{ and } A^- = (\mu_{A^-W}(x_j), v_{A^-W}(x_j)) \quad (19)$$

where,

$$\mu_{A*W}(x_j) = \left( \left( \max_i \mu_{A_iW}(x_j) \middle| j \in J_1 \right), \left( \min_i \mu_{A_iW}(x_j) \middle| j \in J_2 \right) \right) \quad (20)$$

$$v_{A*W}(x_j) = \left( \left( \min_i v_{A_iW}(x_j) \middle| j \in J_1 \right), \left( \max_i v_{A_iW}(x_j) \middle| j \in J_2 \right) \right) \tag{21}$$

$$\mu_{A-W}(x_j) = \left( \left( \min_i \mu_{A_iW}(x_j) \middle| j \in J_1 \right), \left( \max_i \mu_{A_iW}(x_j) \middle| j \in J_2 \right) \right) \tag{22}$$

$$v_{A-W}(x_j) = \left( \left( \max_i v_{A_iW}(x_j) \middle| j \in J_1 \right), \left( \min_i v_{A_iW}(x_j) \middle| j \in J_2 \right) \right) \tag{23}$$

**Step 5.** calculate the separation measures and calculate the relative closeness.

In order to determine the separation between ratings of vendors, a q-ROF distance measure as proposed by Pınar and Boran [11] is used. The separation measures, $S_i^*$ and $S_i^-$, are determined by Equations (24) and (25) respectively.

$$S^* = \sqrt[p]{\frac{1}{2n}\sum_{j=1}^{n}\left\{ \begin{array}{l} \left|(1-k)\left(\mu_{A_iW}(x_j) - \mu_{A*W}(x_j)\right) + k\left(\sqrt[q]{1 - v_{A_iW}^q(x_j)} - \sqrt[q]{1 - v_{A*W}^q(x_j)}\right)\right|^p + \\ \left|(1-k)\left(v_{A_iW}(x_j) - v_{A*W}(x_j)\right) + k\left(\sqrt[q]{1 - \mu_{A_iW}^q(x_j)} - \sqrt[q]{1 - \mu_{A*W}^q(x_j)}\right)\right|^p \end{array} \right\}}$$

$$S^- = \sqrt[p]{\frac{1}{2n}\sum_{j=1}^{n}\left\{ \begin{array}{l} \left|(1-k)\left(\mu_{A_iW}(x_j) - \mu_{A-W}(x_j)\right) + k\left(\sqrt[q]{1 - v_{A_iW}^q(x_j)} - \sqrt[q]{1 - v_{A-W}^q(x_j)}\right)\right|^p + \\ \left|(1-k)\left(v_{A_iW}(x_j) - v_{A-W}(x_j)\right) + k\left(\sqrt[q]{1 - \mu_{A_iW}^q(x_j)} - \sqrt[q]{1 - \mu_{A-W}^q(x_j)}\right)\right|^p \end{array} \right\}}$$

where $p = 1, 2, \ldots, n$ and

$$k = \left( \frac{1}{2}q^2 + \frac{3}{2}q - \frac{1}{3} \right) \middle/ \left( q^2 + 3q + 1 \right), \ k \in \left[ \frac{1}{3}, \frac{1}{2} \right] \tag{24}$$

After separation measures are determined, the relative closeness coefficient ($C_{i*}$) is calculated with Equation (25):

$$C_{i*} = \frac{S_i^-}{S_i^+ + S_i^-} \ where \ 0 \leq C_{i*} \leq 1 \tag{25}$$

**Step 6.** Compare the available vendors and choose the best one.

Descending order of $C_{i*}$'s gives the ranking of alternatives.

## 5. Case Study

A case study is offered to demonstrate the feasibility of combining the IF and VIKOR techniques. As a buyer in this case study, a call center organization has a supplier selection challenge. Alternatives are chosen based on candidates speaking the same language, proposing similar answers, and possessing a comparable level of accuracy.

### 5.1. Application of IF and VIKOR Method

Three DMs who work at the buyer company as a software developer, product owner, and researcher are chosen for our case study and are denoted in the following sections as D, PO, and R. Additionally, DMs are weighted in the case study for their expertise in the purchasing department based on their skills and experiences. The ordered weights of DMs are 0.35, 0.4, and 0.25. Job descriptions and opinion of the decision makers are considered while determining weight of them. The Product Owner carries the most weight, as s/he is the one who is most knowledgeable about the solutions. Following that, developer has the second highest weight due to its superior knowledge of maintenance, adaptability, and security. Finally, due to his or her knowledge of speech recognition technology, the researcher is chosen as the decision-maker with the lowest weight. These DMs assessed four suppliers using the given criteria and sub-criteria. The linguistic terms are listed in Table 3, and the linguistic weights assigned to each criterion are listed in Table 4.

**Table 3.** Linguistic importance alternatives.

| Linguistic Importance Alternatives | IFNs |
|---|---|
| Absolutely Low (AL) | (0.05, 0.95) |
| Low (L) | (0.2, 0.65) |
| Fairly Low (FL) | (0.35, 0.55) |
| Medium (M) | (0.5, 0.5) |
| Fairly High (FH) | (0.65, 0.25) |
| Very High (VH) | (0.8, 0.05) |
| Absolutely High (AH) | (0.9, 0.1) |

**Table 4.** Linguistic criteria significance.

| Linguistic Importance of Criteria | IFNs |
|---|---|
| No influence (N) | (0.15, 0.8) |
| Low influence (L) | (0.2, 0.65) |
| Medium -Low influence (ML) | (0.4, 0.45) |
| Medium influence (M) | (0.5, 0.5) |
| Medium High influence (MH) | (0.55, 0.3) |
| High influence (H) | (0.7, 0.2) |
| Very High influence (VH) | (0.9, 0.1) |

The evaluation results based on the linguistic terms are shown in Tables 5 and 6.

**Table 5.** Information of four alternatives.

| | Decision Makers | Alternative 1 | Alternative 2 | Alternative 3 | Alternative 4 |
|---|---|---|---|---|---|
| | D | AH | AH | VH | FH |
| WER | PO | VH | AH | FH | M |
| | R | AH | AH | FH | M |
| | D | M | VH | VH | VH |
| Technological leadership | PO | VH | AH | AH | M |
| | R | FH | AH | VH | M |
| | D | AH | L | FL | M |
| Unit pricedollar/min | PO | AH | FL | M | FH |
| | R | AH | M | FL | M |
| | D | VH | VH | VH | VH |
| Hardware&software cost | PO | AH | VH | FH | FH |
| | R | VH | AH | AH | VH |
| | D | FH | FH | FL | L |
| Support activities | PO | VH | M | M | FL |
| | R | P | FL | M | M |
| | D | FM | VH | VH | FL |
| Workflow tools | PO | FL | VH | M | M |
| | R | FH | M | M | M |
| | D | AH | AH | AH | FH |
| Num of types of format supporting | PO | VH | AH | FH | FH |
| | R | VH | VH | VH | M |
| | D | AH | VH | FH | FL |
| Capability of handling all accents | PO | VH | AH | M | L |
| | R | AH | FH | FL | FL |
| | D | M | VH | AH | M |
| Multiple languages | PO | FL | FH | AH | M |
| | R | M | FH | AH | FL |
| | D | VH | AH | VH | FH |
| Experienced industries | PO | VH | AH | AH | FH |
| | R | FH | VH | AH | VH |
| | D | AH | VH | VH | VH |
| Integration capability | PO | VH | FH | FH | M |
| | R | VH | VH | FH | M |
| | D | FH | AH | AH | VH |
| Easy deployment | PO | M | AH | AH | AH |
| | R | FL | AH | AH | VH |

**Table 6.** Importance weights of criteria.

|  | **Developer** | **PO** | **Researcher** |
| --- | --- | --- | --- |
| WER | VH | H | VH |
| Technological leadership | MH | H | VH |
| Unit pricedollar/min | N | VH | L |
| Hardware&software cost | H | N | M |
| Support activities | VH | H | ML |
| Workflow tools | MH | H | M |
| Num of types of format supporting | H | M | MH |
| Capability of handling all accents | L | MH | VH |
| Multiple languages | L | H | M |
| Experienced industries | L | VH | H |
| Integration capability | VH | H | M |
| Easy deployment | VH | MH | L |

In first step, IF decision matrix is determined by SIFWA operator and results are shown in Table 7. Following formula is used.

**Table 7.** IF decision matrix and subjective weights of criteria.

|  | **Alternative 1** | **Alternative 2** | **Alternative 3** | **Alternative 4** | **w** |
| --- | --- | --- | --- | --- | --- |
| C1 | (0.86, 0.07) | (0.9, 0.1) | (0.71, 0.15) | (0.55, 0.41) | (0.84, 0.13) |
| C2 | (0.67, 0.19) | (0.87, 0.08) | (0.85, 0.07) | (0.62, 0.26) | (0.72, 0.20) |
| C3 | (0.9, 0.1) | (0.32, 0.57) | (0.41, 0.53) | (0.56, 0.39) | (0.48, 0.44) |
| C4 | (0.85, 0.07) | (0.83, 0.06) | (0.78, 0.12) | (0.75, 0.10) | (0.40, 0.52) |
| C5 | (0.6, 0.2) | (0.52, 0.42) | (0.45, 0.52) | (0.32, 0.57) | (0.73, 0.20) |
| C6 | (0.48, 0.46) | (0.74, 0.10) | (0.62, 0.26) | (0.45, 0.52) | (0.60, 0.30) |
| C7 | (0.84, 0.06) | (0.88, 0.08) | (0.80, 0.13) | (0.61, 0.30) | (0.59, 0.33) |
| C8 | (0.87, 0.07) | (0.82, 0.10) | (0.52, 0.53) | (0.28, 0.56) | (0.54, 0.34) |
| C9 | (0.44, 0.52) | (0.71, 0.15) | (0.90, 0.10) | (0.46, 0.51) | (0.46, 0.42) |
| C10 | (0.77, 0.08) | (0.88, 0.084) | (0.87, 0.08) | (0.69, 0.17) | (0.65, 0.27) |
| C11 | (0.84, 0.06) | (0.75, 0.10) | (0.71, 0.15) | (0.62, 0.26) | (0.75, 0.21) |
| C12 | (0.52, 0.42) | (0.9, 0.1) | (0.90, 0.10) | (0.85, 0.07) | (0.6, 0.28) |

$r_{ij}$ is calculated based on Equation (10) which is explained previous section.

In the formula $i$ represent alternatives, $j$ represents criteria, and $k$ represents DMs.

In second step, normalized decision matrix and criteria weights are calculated and figured in Table 8.

**Table 8.** Normalized decision matrix and criteria weights.

|  | **C1** | **C2** | **C3** | **C4** | **C5** | **C6** | **C7** | **C8** | **C9** | **C10** | **C11** | **C12** |
| --- | --- | --- | --- | --- | --- | --- | --- | --- | --- | --- | --- | --- |
| Alt1 | 0.08 | 0.76 | 0.00 | 0.06 | 0.01 | 0.96 | 0.11 | 0.01 | 1.00 | 0.52 | 0.02 | 1.00 |
| Alt2 | 0.06 | 0.03 | 1.00 | 0.16 | 0.53 | 0.00 | 0.07 | 0.08 | 0.35 | 0.02 | 0.37 | 0.06 |
| Alt3 | 0.47 | 0.08 | 0.86 | 0.72 | 0.80 | 0.45 | 0.30 | 0.77 | 0.00 | 0.04 | 0.56 | 0.06 |
| Alt4 | 0.99 | 1.00 | 0.59 | 0.97 | 1.00 | 1.00 | 0.99 | 1.00 | 0.96 | 1.00 | 1.00 | 0.11 |
| wjS | 0.11 | 0.10 | 0.07 | 0.05 | 0.10 | 0.08 | 0.08 | 0.08 | 0.07 | 0.09 | 0.10 | 0.09 |

Normalized decision matrix is obtained by Equation (27).

$$w_j^s = \frac{u_j + \frac{u_j}{(u_j+v_j)}\pi_{ij}}{\sum_{j=1}^n u_j + \frac{u_j}{(u_j+v_j)}\pi_j} \text{ where, } \pi_j = 1 - u_j - v_j \tag{26}$$

Also, IF best and worst ideal solutions are obtained as shown in Table 8. In case study, while unit price and hardware & software cost criteria are considered as cost, the others are benefit.

**Fj\*** (0.9,0.07)  (0.87,0.07)  (0.9,0.1)  (0.85,0.06)  (0.6,0.2)  (0.74,0.1)  (0.88,0.06)
(0.87,0.07)  (0.9,0.1)  (0.88,0.08)  (0.84,0.06)  (0.9,0.07)

**Fj-** (0.55,0.41)  (0.62,0.26)  (0.32,0.57)  (0.75,0.12)  (0.32,0.57)  (0.45,0.46)  (0.61,0.3)
(0.28,0.56)  (0.44,0.52)  (0.69,0.17)  (0.62,0.26)  (0.52,0.42)

Next, IFE value for all sub-criteria are determined according to objective weighting approach. Results are as shown in Table 9.

$$E_j = -\frac{1}{mln2}\sum_i^m \left(\mu_{ij}ln\mu_{ij} + v_{Aij}lnv_{ij} - (1-\pi_{ij})\ln(1-\pi_{ij}) - \pi_{ij}\ln 2\right) \tag{27}$$

$$w_j^O = \frac{1-E_j}{\sum_{J=1}^n 1 - E_j}, \ 0 \leq w_j^O \leq 1 \text{ and summation of w should be equal to one.} \tag{28}$$

**Table 9.** IFE values and objective weights of criteria.

|      | C1    | C2    | C3    | C4    | C5    | C6    | C7    | C8    | C9    | C10   | C11   | C12   |
|------|-------|-------|-------|-------|-------|-------|-------|-------|-------|-------|-------|-------|
| Ej   | 0.651 | 0.640 | 0.846 | 0.513 | 0.945 | 0.873 | 0.602 | 0.727 | 0.794 | 0.544 | 0.658 | 0.590 |
| WjO  | 0.096 | 0.100 | 0.042 | 0.135 | 0.015 | 0.035 | 0.110 | 0.075 | 0.057 | 0.126 | 0.095 | 0.113 |

Then, weight is calculated by following equation.

$W = w_j^O\theta + (1-\theta)w_j^O$, $\theta$ is decided as 0.5 in the study for simplicity. The effect of changing the value of $\theta$ is analyzed in the sensitivity analysis Section 5.2, shown at Table 10.

**Table 10.** Weighted normalized decision matrix.

|      | C1    | C2    | C3    | C4    | C5    | C6    | C7    | C8    | C9    | C10   | C11   | C12   |
|------|-------|-------|-------|-------|-------|-------|-------|-------|-------|-------|-------|-------|
| Alt1 | 0.008 | 0.076 | 0.000 | 0.006 | 0.001 | 0.057 | 0.011 | 0.001 | 0.062 | 0.055 | 0.002 | 0.100 |
| Alt2 | 0.007 | 0.003 | 0.054 | 0.015 | 0.030 | 0.000 | 0.007 | 0.006 | 0.021 | 0.002 | 0.036 | 0.006 |
| Alt3 | 0.048 | 0.008 | 0.046 | 0.068 | 0.045 | 0.026 | 0.029 | 0.058 | 0.000 | 0.004 | 0.054 | 0.006 |
| Alt4 | 0.101 | 0.099 | 0.032 | 0.092 | 0.057 | 0.059 | 0.094 | 0.076 | 0.059 | 0.107 | 0.096 | 0.011 |

Group utility and individual regret is calculated by following equation. Si, Ri and Qi are calculated based on the Equations (12)–(14), shown at Tables 11 and 12.

**Table 11.** Group utility and individual regret.

|               | Si       | Ri       | Qi       |
|---------------|----------|----------|----------|
| **Alternative 1** | 0.376271 | 0.099785 | 0.572289 |
| **Alternative 2** | 0.186553 | 0.053566 | 0        |
| **Alternative 3** | 0.392623 | 0.067966 | 0.284011 |
| **Alternative 4** | 0.881708 | 0.10659  | 1        |

**Table 12.** Sensitivity Analysis with different values of $\theta$.

| Alternative/Qi | $\Theta = 0$ | $\Theta = 0.25$ | $\Theta = 0.5$ | $\Theta = 0.75$ | $\Theta = 1$ |
|----------------|----------|----------|----------|----------|----------|
| Alternative 1  | 0.871664 | 0.721977 | 0.572289 | 0.422602 | 0.272914 |
| Alternative 2  | 0        | 0        | 0        | 0        | 0        |
| Alternative 3  | 0.271584 | 0.277798 | 0.284011 | 0.290224 | 0.296438 |
| Alternative 4  | 1        | 1        | 1        | 1        | 1        |

Condition 1: Acceptable advantage

Alternative 2 has the best Qi value with 0 and alternative 3 is the second-best. Therefore, DQ = 1/(4 − 1) = 0.33 and Q3 − Q2 = 0.28 − 0 = 0.28, 0.42 < 0.33 so it does not provide acceptable advantage condition.

When looking at the third-best which is alternative 1, Q1 − Q2 = 0.57 − 0 = 0.57 so it provides the condition.

Condition 2: Sustainability acceptance in result

Alternative 2 has the best individual regret value, too.

As a result, alternative 2 is obtained as the best alternative in this study with providing two acceptable conditions of VIKOR method.

### 5.2. Sensitivity Analysis for IF-VIKOR Method

Because the study considers both collective utility and individual regret to be equally important, is defined as 0.5. A ratio greater than 0.5 indicates that collective utility is more important, whereas a value less than 0.5 indicates that individual regret is more essential. Although the value is frequently expressed as 0.5 in the literature, it may also be expressed as 0.25 or 0.75. As a result, this section of the study examines the effects of various parameter values.

For $\theta$ is equal to 0;

$Q3 - Q2 = 0.271584 - 0 = 0.272$, which is smaller than DQ. So, $Q1-Q2 = 0.872-0 = 0.872$, which is greater than DQ. Therefore, solution is defined as alternative 2 and alternative 1. It means that the DMs can select any one of them. Moreover, still alternative 2 is best for individual regret and sustainable acceptance requirement is met, too.

For $\theta$ is equal to 0.25;

$Q3 - Q2 = 0.28 - 0 = 0.28$ and $0.28 < 0.33$, so continue to next step that is $Q1-Q2$. The difference is equal to 0.72 that means it met the condition. Therefore, DMs can select either alternative 2 or alternative 1.

For $\theta$ is equal to 0.75;

$Q3 - Q2 = 0.29 - 0 = 0.29$ and $0.29 < 0.33$, so continue to next step that is $Q1-Q2$. The difference is equal to 0.42 that means it met the condition. Therefore, DMs can select either alternative 2 or alternative 1.

For $\theta$ is equal to 1;

$Q1 - Q2 = 0.27 - 0 = 0.27$, which is smaller than DQ. In second step $Q3-Q2 = 0.3$ which is again smaller than 0.33. So, move to the third step i.e., $Q4 - Q2 = 1$, the value of which is greater than DQ. Therefore, with this value DMs cans select alternative 2 or alternative 4.

As a result, calculations show that alternative 2 is best for all values of the parameter $\theta$, but second option is changed when $\theta$ is equal to 1. It means, if only group utility is considered as important, then alternatives can change.

### 5.3. Application of q-ROF TOPSIS Method

Our second method q-ROF TOPSIS is applied to the same problem, with the DMs weights assumed to be [0.35, 0.4, 0.25] as in IF-VIKOR method. In q-ROF TOPSIS nine-level scale is used for linguistic terms of criteria (Table 13) and alternatives (Table 14).

**Table 13.** Linguistic terms for criteria ratings.

| Linguistic Terms | μ | v |
|---|---|---|
| Absolutely High (AH) | 0.95 | 0.15 |
| Very High (VH) | 0.85 | 0.25 |
| High (H) | 0.75 | 0.35 |
| Fairly High (FH) | 0.65 | 0.45 |
| Medium (M) | 0.55 | 0.55 |
| Medium Low (ML) | 0.45 | 0.65 |
| Fairly Low (FL) | 0.35 | 0.75 |
| Low (L) | 0.25 | 0.85 |
| Absolutely Low (AL) | 0.15 | 0.95 |

**Table 14.** Linguistic terms for alternative ratings.

| Linguistic Terms | μ | v |
|---|---|---|
| Extremely High Influence (EH) | 0.95 | 0.15 |
| Very High influence (VH) | 0.85 | 0.25 |
| High influence (H) | 0.75 | 0.35 |
| Medium High influence (MH) | 0.65 | 0.45 |
| Medium influence (M) | 0.55 | 0.55 |
| Medium Low (ML) | 0.45 | 0.65 |
| Low influence (L) | 0.35 | 0.75 |
| Very Low influence (VL) | 0.25 | 0.85 |
| No influence (N) | 0.15 | 0.95 |

**Step 1.** Aggregate the DMs ratings.

DMs evaluations in linguistic terms are converted to q-ROFNs with the help of Table 14. The ratings of the alternatives in q-ROFNs are shown in Table 15.

**Table 15.** DMs ratings for alternatives in q-ROFNs, *(a)* DM1, *(b)* DM2, *(c)* DM3.

| DM1 | A1 | | A2 | | A3 | | A4 | |
|---|---|---|---|---|---|---|---|---|
| | μ | v | μ | v | μ | v | μ | v |
| C1 | 0.95 | 0.15 | 0.95 | 0.15 | 0.85 | 0.25 | 0.65 | 0.45 |
| C2 | 0.55 | 0.55 | 0.85 | 0.25 | 0.85 | 0.25 | 0.85 | 0.25 |
| C3 | 0.95 | 0.15 | 0.25 | 0.85 | 0.35 | 0.75 | 0.55 | 0.55 |
| C4 | 0.85 | 0.25 | 0.85 | 0.25 | 0.85 | 0.25 | 0.85 | 0.25 |
| C5 | 0.65 | 0.45 | 0.65 | 0.45 | 0.35 | 0.75 | 0.25 | 0.85 |
| C6 | 0.55 | 0.55 | 0.85 | 0.25 | 0.85 | 0.25 | 0.35 | 0.75 |
| C7 | 0.95 | 0.15 | 0.95 | 0.15 | 0.95 | 0.15 | 0.65 | 0.45 |
| C8 | 0.95 | 0.15 | 0.85 | 0.25 | 0.65 | 0.45 | 0.35 | 0.75 |
| C9 | 0.55 | 0.55 | 0.85 | 0.25 | 0.95 | 0.15 | 0.55 | 0.55 |
| C10 | 0.85 | 0.25 | 0.95 | 0.15 | 0.85 | 0.25 | 0.65 | 0.45 |
| C11 | 0.95 | 0.15 | 0.85 | 0.25 | 0.85 | 0.25 | 0.85 | 0.25 |
| C12 | 0.65 | 0.45 | 0.95 | 0.15 | 0.95 | 0.15 | 0.85 | 0.25 |

| DM2 | A1 | | A2 | | A3 | | A4 | |
|---|---|---|---|---|---|---|---|---|
| | μ | v | μ | v | μ | v | μ | v |
| C1 | 0.85 | 0.25 | 0.95 | 0.15 | 0.65 | 0.45 | 0.55 | 0.55 |
| C2 | 0.85 | 0.25 | 0.95 | 0.15 | 0.95 | 0.15 | 0.55 | 0.55 |
| C3 | 0.95 | 0.15 | 0.35 | 0.75 | 0.55 | 0.55 | 0.65 | 0.45 |
| C4 | 0.95 | 0.15 | 0.85 | 0.25 | 0.65 | 0.45 | 0.65 | 0.45 |
| C5 | 0.85 | 0.25 | 0.55 | 0.55 | 0.55 | 0.55 | 0.35 | 0.75 |
| C6 | 0.35 | 0.75 | 0.85 | 0.25 | 0.55 | 0.55 | 0.55 | 0.55 |
| C7 | 0.85 | 0.25 | 0.95 | 0.15 | 0.65 | 0.45 | 0.65 | 0.45 |
| C8 | 0.85 | 0.25 | 0.95 | 0.15 | 0.55 | 0.55 | 0.25 | 0.85 |
| C9 | 0.35 | 0.75 | 0.65 | 0.45 | 0.95 | 0.15 | 0.55 | 0.55 |
| C10 | 0.85 | 0.25 | 0.95 | 0.15 | 0.95 | 0.15 | 0.65 | 0.45 |
| C11 | 0.85 | 0.25 | 0.65 | 0.45 | 0.65 | 0.45 | 0.55 | 0.55 |
| C12 | 0.55 | 0.55 | 0.95 | 0.15 | 0.95 | 0.15 | 0.95 | 0.15 |

| DM3 | A1 | | A2 | | A3 | | A4 | |
|---|---|---|---|---|---|---|---|---|
| | μ | v | μ | v | μ | v | μ | v |
| X1 | 0.95 | 0.15 | 0.95 | 0.15 | 0.65 | 0.45 | 0.55 | 0.55 |
| X2 | 0.65 | 0.45 | 0.95 | 0.15 | 0.85 | 0.25 | 0.55 | 0.55 |
| X3 | 0.95 | 0.15 | 0.55 | 0.55 | 0.35 | 0.75 | 0.55 | 0.55 |
| X4 | 0.85 | 0.25 | 0.95 | 0.15 | 0.95 | 0.15 | 0.85 | 0.25 |
| X5 | 0.25 | 0.85 | 0.35 | 0.75 | 0.55 | 0.55 | 0.55 | 0.55 |
| X6 | 0.65 | 0.45 | 0.55 | 0.55 | 0.55 | 0.55 | 0.55 | 0.55 |
| X7 | 0.85 | 0.25 | 0.85 | 0.25 | 0.85 | 0.25 | 0.55 | 0.55 |
| X8 | 0.95 | 0.15 | 0.65 | 0.45 | 0.35 | 0.75 | 0.35 | 0.75 |
| X9 | 0.55 | 0.55 | 0.65 | 0.45 | 0.95 | 0.15 | 0.35 | 0.75 |
| X10 | 0.65 | 0.45 | 0.85 | 0.25 | 0.95 | 0.15 | 0.85 | 0.25 |
| X11 | 0.85 | 0.25 | 0.85 | 0.25 | 0.65 | 0.45 | 0.55 | 0.55 |
| X12 | 0.35 | 0.75 | 0.95 | 0.15 | 0.95 | 0.15 | 0.85 | 0.25 |

These q-ROFNs are aggregated with DM weights with q-ROFWA operator given in Equation (16). We get an aggregated q-ROF decision matrix, as below.

$$
R=\begin{array}{c|cccccc}
 & C1 & C2 & C3 & C4 & C5 & C6 \\
\hline
A1 & (0.924;0.184;0.591) & (0.742;0.382;0.812) & (0.950;0.150;0.518) & (0.905;0.204;0.631) & (0.732;0.417;0.812) & (0.530;0.592;0.863) \\
A2 & (0.950;0.150;0.518) & (0.928;0.179;0.581) & (0.405;0.725;0.821) & (0.888;0.220;0.662) & (0.563;0.554;0.867) & (0.810;0.304;0.760) \\
A3 & (0.748;0.366;0.811) & (0.905;0.204;0.631) & (0.455;0.662;0.850) & (0.849;0.278;0.715) & (0.500;0.613;0.864) & (0.714;0.417;0.826) \\
A4 & (0.591;0.513;0.870) & (0.714;0.417;0.826) & (0.596;0.508;0.870) & (0.795;0.316;0.775) & (0.405;0.725;0.821) & (0.500;0.613;0.864) \\
 & C7 & C8 & C9 & C10 & C11 & C12 \\
A1 & (0.899;0.209;0.641) & (0.924;0.184;0.591) & (0.492;0.623;0.862) & (0.818;0.290;0.753) & (0.899;0.209;0.641) & (0.563;0.554;0.867) \\
A2 & (0.935;0.170;0.562) & (0.886;0.236;0.662) & (0.748;0.366;0.811) & (0.935;0.170;0.562) & (0.795;0.316;0.775) & (0.950;0.150;0.518) \\
A3 & (0.866;0.264;0.693) & (0.563;0.554;0.867) & (0.950;0.150;0.518) & (0.928;0.179;0.581) & (0.748;0.366;0.811) & (0.950;0.150;0.518) \\
A4 & (0.629;0.473;0.864) & (0.318;0.789;0.782) & (0.516;0.594;0.868) & (0.725;0.389;0.825) & (0.714;0.417;0.826) & (0.905;0.204;0.631) \\
\end{array}
$$

**Step 2.** Calculate the importance weights of the criteria.

DMs ratings in linguistic terms for criteria weights are converted into q-ROFNs and aggregated with the help of Equation (17). 12 Criteria weights are calculated as $[w_1 = 0.108, w_2 = 0.097, w_3 = 0.057, w_4 = 0.059, w_5 = 0.093, w_6 = 0.086, w_7 = 0.085, w_8 = 0.078, w_9 = 0.071, w_{10} = 0.084, w_{11} = 0.097, w_{12} = 0.084]$

**Step 3.** Set up a weighted decision matrix.

Weighted aggregated q-ROF decision matrix is determined as below:

$$
R'=\begin{array}{c|cccccc}
 & C1 & C2 & C3 & C4 & C5 & C6 \\
\hline
A1 & (0.536;0.833;0.645) & (0.368;0.910;0.581) & (0.473;0.897;0.557) & (0.425;0.911;0.553) & (0.356;0.922;0.556) & (0.240;0.956;0.483) \\
A2 & (0.575;0.814;0.646) & (0.525;0.846;0.630) & (0.158;0.982;0.368) & (0.409;0.915;0.550) & (0.263;0.946;0.512) & (0.398;0.903;0.586) \\
A3 & (0.385;0.897;0.605) & (0.498;0.856;0.629) & (0.178;0.977;0.397) & (0.379;0.927;0.529) & (0.231;0.955;0.487) & (0.336;0.928;0.547) \\
A4 & (0.291;0.930;0.554) & (0.350;0.918;0.567) & (0.238;0.962;0.459) & (0.343;0.934;0.524) & (0.185;0.971;0.430) & (0.225;0.959;0.475) \\
 & C7 & C8 & C9 & C10 & C11 & C12 \\
A1 & (0.471;0.876;0.607) & (0.484;0.877;0.597) & (0.208;0.967;0.444) & (0.402;0.901;0.589) & (0.491;0.860;0.627) & (0.254;0.951;0.496) \\
A2 & (0.512;0.861;0.611) & (0.446;0.894;0.582) & (0.336;0.931;0.538) & (0.511;0.861;0.610) & (0.403;0.895;0.602) & (0.533;0.852;0.612) \\
A3 & (0.440;0.893;0.587) & (0.247;0.955;0.484) & (0.506;0.874;0.589) & (0.502;0.865;0.609) & (0.371;0.907;0.586) & (0.533;0.852;0.612) \\
A4 & (0.288;0.939;0.530) & (0.136;0.982;0.372) & (0.219;0.964;0.456) & (0.341;0.923;0.557) & (0.350;0.919;0.566) & (0.476;0.875;0.607) \\
\end{array}
$$

**Step 4.** Determine the Positive and Negative Ideal Solutions:

All criteria other than C3 (Unit price dollar/min) and C4 (Hardware & software cost) are benefit criteria. This stage involves determining the intuitionistic fuzzy positive ideal solution IFPIS and the intuitionistic fuzzy negative ideal solution IFNIS. Assume that J1 and J2 are benefit and cost criteria, respectively. So, with the help of Equations (19)–(23) the q-ROFPIS ($A^*$) and the q-ROFNIS ($A^-$) were calculated as follows:

$$
A* = \begin{array}{c}
C_1 \\ C_2 \\ C_3 \\ C_4 \\ C_5 \\ C_6 \\ C_7 \\ C_8 \\ C_9 \\ C_{10} \\ C_{11} \\ C_{12}
\end{array}
\begin{bmatrix}
(0.575, 0.814, 0.646) \\
(0.525, 0.846, 0.630) \\
(0.158, 0.982, 0.368) \\
(0.343, 0.934, 0.524) \\
(0.356, 0.922, 0.556) \\
(0.398, 0.903, 0.586) \\
(0.512, 0.861, 0.611) \\
(0.484, 0.877, 0.597) \\
(0.506, 0.874, 0.584) \\
(0.511, 0.861, 0.610) \\
(0.491, 0.860, 0.627) \\
(0.533, 0.852, 0.612)
\end{bmatrix}
\qquad
A^- = \begin{array}{c}
C_1 \\ C_2 \\ C_3 \\ C_4 \\ C_5 \\ C_6 \\ C_7 \\ C_8 \\ C_9 \\ C_{10} \\ C_{11} \\ C_{12}
\end{array}
\begin{bmatrix}
(0.291, 0.930, 0.554) \\
(0.350, 0.918, 0.567) \\
(0.473, 0.897, 0.557) \\
(0.425, 0.911, 0.553) \\
(0.185, 0.971, 0.430) \\
(0.225, 0.959, 0.475) \\
(0.288, 0.939, 0.530) \\
(0.136, 0.982, 0.372) \\
(0.208, 0.967, 0.444) \\
(0.341, 0.923, 0.557) \\
(0.350, 0.919, 0.566) \\
(0.254, 0.951, 0.496)
\end{bmatrix}
$$

**Step 5.** Determine the separation measures.

The separation measures, $S_i^*$ and $S_i^-$, are determined by Equation (24) with the help of the distance measure proposed by Pinar and Boran [11] and given as Table 16:

**Table 16.** Separation measures.

| Alternatives | $S^*$ | $S^-$ | $Ci^*$ |
|---|---|---|---|
| A1 | 0.071 | 0.057 | 0.448 |
| A2 | 0.021 | 0.107 | 0.833 |
| A3 | 0.043 | 0.085 | 0.661 |
| A4 | 0.102 | 0.026 | 0.201 |

*5.4. Sensitivity Analyses in q-ROF TOPSIS*

A sensitivity analysis is performed to see the effects of parameter *q* in q-ROF TOPSIS. For q = (2, 3, 4, 5, 6, 7, 8, 9, 10) the values of closeness coefficient for vendors are presented in Table 17. It can be easily seen that the increase in q values does not change the rank of the alternatives. We take p = 1 and q = 3 which are the most stable parameters for this method [11].

**Table 17.** q Parameter analyses.

| *q Values* \ **Alternatives** | 2 | 3 | 4 | 5 | 6 | 7 | 8 | 9 | 10 |
|---|---|---|---|---|---|---|---|---|---|
| A1 | 0.445 | 0.448 | 0.450 | 0.452 | 0.454 | 0.456 | 0.458 | 0.460 | 0.462 |
| A2 | 0.829 | 0.833 | 0.836 | 0.837 | 0.839 | 0.840 | 0.841 | 0.841 | 0.842 |
| A3 | 0.649 | 0.661 | 0.668 | 0.673 | 0.677 | 0.680 | 0.682 | 0.684 | 0.686 |
| A4 | 0.197 | 0.201 | 0.203 | 0.204 | 0.205 | 0.205 | 0.205 | 0.206 | 0.206 |

As seen in Figure 4, the outcomes of both approaches are consistent, with the exception of =1 in the IF-VIKOR method, which favors Alternative 1 over Alternative 3. When we compare the two approaches, we see that whereas IF-VIKOR produces varied rankings due to its structure containing group utility and regret parameters, the q-ROF TOPSIS ranking results remain constant regardless of the q parameter's value. A nine-level scale facilitates the conversion of language concepts to q-ROF values, therefore quantifying verbal assessment and decreasing uncertainty. IF-VIKOR also generates extreme values of 0 and 1, but q-ROF TOPSIS produces moderate values.

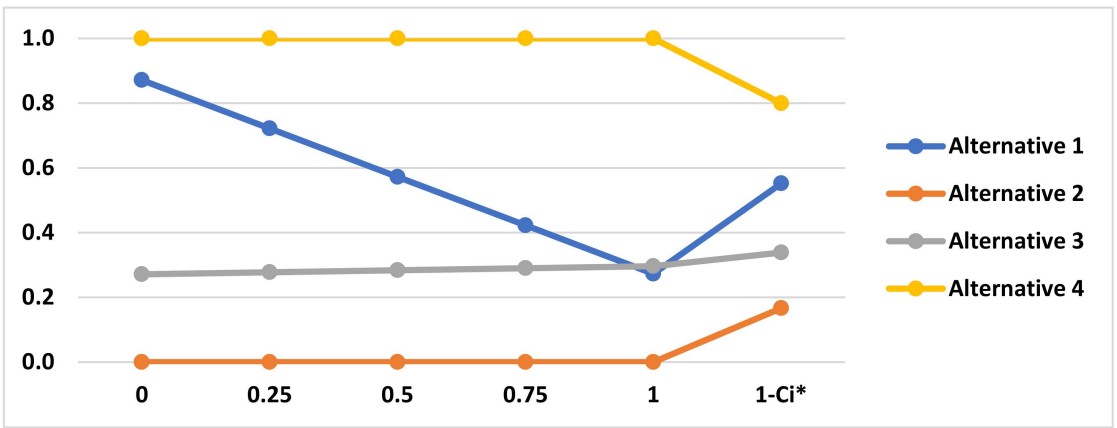

**Figure 4.** Comparison of IF-VIKOR and q-ROF TOPSIS methods.

## 6. Results and Discussion

In this study, we use two different methodologies, IF-VIKOR and q-ROF TOPSIS for speech recognition software supplier selection. We compare these methods in Figure 4

and the results are given in Table 18. To make the results q-ROF TOPSIS compatible with IF-VIKOR methods', Ci* values are normalized as 1-Ci*.

**Table 18.** Results of IF-VIKOR and q-ROF TOPSIS methods.

| | IF-VIKOR θ Value | | | | | q-ROF TOPSIS |
| | 0 | 0.25 | 0.5 | 0.75 | 1 | 1-Ci* |
|---|---|---|---|---|---|---|
| Alternative 1 | 0.872 | 0.722 | 0.572 | 0.423 | 0.273 | 0.552 |
| Alternative 2 | 0.000 | 0.000 | 0.000 | 0.000 | 0.000 | 0.167 |
| Alternative 3 | 0.272 | 0.278 | 0.284 | 0.290 | 0.296 | 0.339 |
| Alternative 4 | 1.000 | 1.000 | 1.000 | 1.000 | 1.000 | 0.799 |

## 7. Conclusions

Organizations should automate their operations in today's market, since competitiveness is more difficult than ever. Automation procedures may be implemented effectively and simply with the help of modern technologies such as machine learning and artificial intelligence. Remote communication between consumers and suppliers, in particular, is critical in light of unpredictable disruptive occurrences such as Covid-19. Additionally, selecting the best supplier has been a major concern for companies and researchers for years. While selecting a supplier for important processes typically requires passing many criteria, many reputable suppliers may meet multiple criteria to differing degrees in today's reality. At this stage, the problem transforms into one of MCDM in an uncertain environment. When studying such a challenge, a company in the customer interaction industry is chosen. The firm develops products such as bots and interactive voice response systems and requires Speech Recognition technology for its mobile applications, which it does not control. The problem is described as supplier selection in this study, and the four most often used primary criteria and twelve sub-criteria are chosen. The DMs rated four suppliers in issue structuring based on their experiences and certain test findings. A hybrid IF-VIKOR and q-ROF TOPSIS technique is used in the decision model. IF enables the evaluation of four possibilities in a fuzzy environment and the ranking of the alternatives according to the decision makers' preferences. As a result, subjectivity and ambiguity are eliminated. Following that, VIKOR evaluates the options using group utility and individual regret levels. After analyzing the issue description, a case study is conducted. As a consequence of the IF-VIKOR approach used in this section, Alternative 2 is determined to be the most suited option, with the highest Qi value and least individual regret. It provides a method for vendors to assess while taking into account the pros and downsides of many criteria. As a second technique, the identical problem is solved using the q-ROF TOPSIS method. The rankings of alternatives produced by q-ROF TOPSIS are similarly consistent with those produced by IF-VIKOR, since it produces steady and moderate results, whereas IF-VIKOR produces extreme outcomes. As the purpose of this study is to fill a vacuum in the literature about the evaluation of voice recognition technologies using both quantifiable and non-quantifiable criteria, both techniques support both of these criteria kinds. A nine-level scale, particularly in q-ROF TOPSIS, is used to transform linguistic concepts to q-ROF numbers, quantifies verbal utterances, and reduces ambiguity.

Other Fuzzy MCDM methodologies may be applied to the selection of IT suppliers in the future. Additionally, precise security requirements might be included in the problem definition, since information security becomes increasingly critical as big data and cloud applications become more prevalent.

## 8. Discussions and Future Research

The study attempted to bridge a gap in the literature by analyzing multiple types of criteria simultaneously. After a variety of approaches is applied, alternative 2 is determined to be the best choice. According to the majority of decision makers, the outcome is appropriate for reality because the alternative has the highest quality level and technological leadership. In the future, the problem could be expanded to include primary security

criteria, as security is becoming increasingly important as personal information becomes more easily accessible. Securing personal information is a critical and challenging issue for all suppliers. Additionally, protecting the solution from cyber-attacks is a critical security problem to provide uninterrupted service. Additionally, limitations can be expanded in future study to bring them closer to real-world issues.

**Author Contributions:** Conceptualization, A.T. and B.D.R. methodology, validation, writing, A.Ü.; investigation, and; resources, data curation, S.P.—review, and editing. All authors have read and agreed to the published version of the manuscript.

**Funding:** This research received no external funding.

**Conflicts of Interest:** The authors declare no conflict of interest. The funders had no role in the design of the study; in the collection, analyses, or interpretation of data; in the writing of the manuscript, or in the decision to publish the results.

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
