# Peer review of "Selection of Suppliers for Speech Recognition Products in IT Projects by Combining Techniques with an Integrated Fuzzy MCDM"

_sustainability, doi:10.3390/su14031777_

Round 1

Reviewer 1 Report

Comments to Authors:

1. Introduction

(1) Page 2: In this section, the manuscript stated that VIKOR was used because the choice includes qualitative and quantitative data. However, is there any literature to support this choice? In addition, the reasons for this choice are not clearly expressed. Is there other MCDM approach for this issue?

  1. Literature Review

(1) Page 3: Wu, Zhou, ChenHuayou? What are these? Are they the citations?

(2) Page 3-6: I'm not quite sure what Tables 1 and 2 really want to express? In addition, it took up a lot of space.

(3) The manuscript reviewed the previous literatures, however, it only listed the previous relevant literatures, and did not review the literature. At the same time, there is no clear expression that the manuscript really wants to fill the gaps in previous researches. Therefore, I strongly recommend that the authors rewrite the literature review section to make it more logical.

  1. Structuring the problem

(1) Page 7: How to decide the sub criteria in Figure 1? Is there any references to support this selection?

(2) In addition, the literature review mainly discusses the methodology applied in relevant issues, but the selection of criteria was not discussed in the literature review.

  1. Constructing the decision model

(1) Page 9: The manuscript used Yager (2014) as the citation style in this section. But in the previous section, the manuscript used the different style, Y.Chen, Wang 2009. It is recommended that the manuscript should use a uniform reference format.

(2) Which criteria select the normalization formula of large attribute?

(3) Which method is used to calculate the weights of the criteria?

  1. Case study

(1) Page 13: How to decide the weights of DMs as 0.35, 0.4 and 0.25?

(2) Page 19: How to determine the Positive and Negative Ideal Solution? What are the Positive and Negative Ideal Solution?

(3) According to my understanding, VIKOR is mainly based on the distance from the positive ideal solution, while TOPSIS is based on the distance from the positive and negative ideal solution. How to combine these two methods? What is the basis of the combination? What is the internal logic?

  1. Results and discussion

(1) Page 21: The manuscript mentioned that Alternative 1 over Alternative 3. Is this conclusion proven in reality?

Author Response

Answer to Reviewers' Comments

Sustainability

Ms. Ref. No.: Sustainability-1542637

Title of Paper:

Selection of suppliers for speech recognition products in IT projects by combining techniques with an integrated Fuzzy MCDM

Thanks for your helpful observations. The actual comments of the reviewer are in italicized font and our answers are in in Red font. Corrections have been made in the text. Language edition completely has been made.

Reviewer 2 Report

Review sustainability-1542637, entitled "Selection of suppliers for speech recognition products in IT projects by combining techniques with an integrated Fuzzy MCDM."

Frist, rewrite the title as: Selection of Suppliers for Speech Recognition Products in IT Projects Using an Integrated Fuzzy MCDM

  • In introduction section, second page better the idea get supported by citing some papers. Justifying the research gap is matter which needs more elaborations. Besides, if we are after the novelty of this work I should say is unjustifiable as we could find many application of fuzzy MCDM in different sectors. As novelty lies in body of knowledge/ methodology, then I agree most with first statement and criteria used for evaluations, but second explanation about contributions is not justifiable as all the fuzzy techniques aimed to do so.
  • In introduction section, many abbreviations exist that first must be written in full phrase then in capital letter. Like q-ROF as author wrote in full in page 9 but it needs to be written in full in this section.
  • At the end of introduction, correct the word “reminder”
  • In literature review and summarized table, it is good idea you add the advantages of your work comparing with others too.
  • Please justify the equations number.
  • Please explains to readers why in Table 3 and 4 different linguistics were used.
  • Before conclusions, managerial implications of work is missing.

Author Response

(The authors gave the same response as above.)

Reviewer 3 Report

This paper proposes two hybrid strategies to determine the ideal speech recognition supplier, and illustrates the effectiveness of the proposed method through specific cases. The research questions in this paper are of practical significance and the research methods are operable. However, the following problems still exist:

  1. There are too few introductions about research contributions in the introduction, and it does not highlight the innovative points of the research in this article. The contributions proposed are also concentrated on the micro-method level. You can combine the problems and add the macro-level contributions.
  2. The introduction says that "quality, affordability, maintenance and adaptability are the focus of the inspection", but it does not explain the reasons for choosing these inspection indicators. The reasons should be pointed out in the introduction and can be described briefly.
  3. The literature review about suppliers Selection and fuzzy MCDM should be further reviewed, and the following works relted your paper should be discussed,  A combined fuzzy DEMATEL and TOPSIS approach for estimating participants in knowledge-intensive crowdsourcing; The partner selection modes for knowledge-based innovation networks: A multiagent simulation
  4. There are too few research introductions about speech recognition vendors in the literature review. This is the problem studied in this article, and the related research introduction must be substantial.
  5. In the third part of problem construction, before introducing the model construction, the problems to be studied in this article should be discussed in detail, and then the model should be introduced on this basis.
  6. In the conclusion part, there are too few prospects for future research. At least two points are mentioned.

Based on the above comments, it is recommended to review carefully after careful revision.

Author Response

(The authors gave the same response as above.)

Reviewer 4 Report

I am pleased to have the opportunity to review this research paper. This study attempted to explore the Selection of suppliers for speech recognition products in IT projects by combining techniques with an integrated Fuzzy MCDM. Although the topic of this research study is interesting and fits within the journal scope, I think authors should apply the comments indicated below to increase the quality of research justification, contributions and findings. The manuscript know lacks in scientific style and structure.

First of all, paper research gap. Please improve this part in introduction section. Introduction is very general and lacked alignment to the research findings, no discussion was provided to derive the implication from. Theoretical and pragmatics implication are vague and need to be better aligned with this paper theoretical underpinnings and proposed process. Furthermore, there is insufficient support and weak arguments in support of the objective that is proposed as well as the model developed. In the final part of the introduction the objectives proposed, originality and gap that would be better covered. Also how the author will perform the methodology.

the topic of this research study is interesting and fits within the journal scope, I think authors should apply the comments indicated to increase the quality of research justification, contributions and findings

What is the originality of this research?  Paper research gap and originality should be better presented at the end of introduction section

Please consider this structure for manuscript final part.

-Discussion

-Conclusion

-Managerial Implication

-Practical/Social Implications

-Discussion needs to be a coherent and cohesive set of arguments that take us beyond this study in particular, and help us see the relevance of what authors have proposed. Authors should create an independent “Discussion” section. Author need to contextualize the findings in the literature, and need to be explicit about the added value of your study towards that literature. Also other studies should be cited to increase the theoretical background of each of the method used. Findings should be contextualized in the literature and should be explicit about the added value of the study towards the literature. Limitations and future research

Questions to be answered:

What practical/professional and academic consequences will this study have for the future of scientific literature (theoretical contributions)?

Why is this study necessary? should make clear arguments to explain what is the originality and value of the proposed model. This should be stated in the final paragraphs of introduction and conclusion sections.

Author Response

REv

Round 2

Reviewer 1 Report

The author was able to answer my questions and revised them in the manuscript.
In addition, it is suggested that the authors should read through the full text, to ensure the correctness of the manuscript.

Author Response

Dear 

In General, a language chack is performed. 

Kind Regards

Reviewer 3 Report

I am ok with this revision

Author Response

(The authors gave the same response as above.)

Reviewer 4 Report

your work is now much better, congratulations. However, for it to be published, it is better that you better support your gap with literature.

Author Response

(The authors gave the same response as above.)

Round 3

Reviewer 4 Report

congratulations, the work is already much better, it is better that you better support your gap with literature.

Author Response

(The authors gave the same response as above.)
